# On clustering network-valued data

**Soumendu Sundar Mukherjee**
Department of Statistics
University of California, Berkeley
Berkeley, California 94720, USA
soumendu@berkeley.edu

**Purnamrita Sarkar**
Department of Statistics and Data Sciences
University of Texas, Austin
Austin, Texas 78712, USA
purna.sarkar@austin.utexas.edu

**Lizhen Lin**
Department of Applied and Computational Mathematics and Statistics
Univeristy of Notre Dame
Notre Dame, Indiana 46556, USA
lizhen.lin@nd.edu

## Abstract

Community detection, which focuses on clustering nodes or detecting communities in (mostly) a single network, is a problem of considerable practical interest and has received a great deal of attention in the research community. While being able to cluster within a network is important, there are emerging needs to be able to *cluster multiple networks*. This is largely motivated by the routine collection of network data that are generated from potentially different populations. These networks may or may not have node correspondence. When node correspondence is present, we cluster networks by summarizing a network by its graphon estimate, whereas when node correspondence is not present, we propose a novel solution for clustering such networks by associating a computationally feasible feature vector to each network based on trace of powers of the adjacency matrix. We illustrate our methods using both simulated and real data sets, and theoretical justifications are provided in terms of consistency.

## 1 Introduction

A network, which is used to model interactions or communications among a set of agents or nodes, is arguably among one of the most common and important representations for modern complex data. Networks are ubiquitous in many scientific fields, ranging from computer networks, brain networks and biological networks, to social networks, co-authorship networks and many more. Over the past few decades, great advancement has been made in developing models and methodologies for inference of networks. There are a range of probabilistic models for networks, starting from the relatively simple Erdös-Rényi model [12], stochastic blockmodels and their extensions [15, 17, 6], to infinite dimensional graphons [28, 13]. These models are often used for community detection, i.e. clustering the nodes in a network. Various community detection algorithms or methods have been proposed, including modularity-based methods [21], spectral methods [25], likelihood-based methods [8, 11, 7, 4], and optimization-based approaches like those based on semidefinite programming [5], etc.

The majority of the work in the community detection literature including the above mentioned ones focus on finding communities among the nodes in *a single network*. While this is still a very important problem with many open questions, there is an emerging need to be able to detect *clusters among multiple network-valued objects*, where a network itself is a

fundamental unit of data. This is largely motivated by the routine collection of populations or subpopulations of network-valued data objects. Technological advancement and the explosion of complex data in many domains has made this a somewhat common practice.

There has been some notable work on graph kernels in the Computer Science literature [27, 26]. In these works the goal is to efficiently compute different types of kernel based similarity measures (or their approximations) between networks. In contrast, we ask the following statistical questions. Can we cluster networks *consistently* from a mixture of graphons, when 1) there is node correspondence and 2) when there isn't? The first situation arises, for example, when one has a network evolving over time, or multiple instances of a network between well-defined objects. If one thinks of them as random samples from a mixture of graphons, then can we cluster them? We propose a simple and general algorithm to address this question, which operates by first obtaining a graphon estimate of each of the networks, constructing a distance matrix between those graphon estimates, and then performing spectral clustering on the distance matrix. We call this algorithm Network Clustering based on Graphon Estimates (NCGE).

The second situation arises when one is interested in global properties of a network. This setting is closer to that of graph kernels. Say we have co-authorship networks from Computer Science and High Energy Physics. Are these different types of networks? There has been a lot of empirical and algorithmic work on featurizing networks or computing kernels between networks. But most of these features require expensive computation. We propose a simple feature based on traces of powers of the adjacency matrix for this purpose which is very cheap to compute as it involves only matrix multiplication. We cluster the networks based on these features and call this method Network Clustering based on Log Moments (NCLM).

We provide some theoretical guarantees for our algorithms in terms of consistency, in addition to extensive simulations and real data examples. The simulation results show that NCGE clearly outperform the naive yet popular method of clustering (vectorized) adjacency matrices in various settings. We also show that, in absence of node correspondence, NCLM is consistently better and faster than methods which featurize networks with different global statistics and graphlet kernels. We also apply NCLM to separate out a mixed bag of real world networks, like co-authorship networks form different domains and ego networks.

The rest of the paper is organized as follows. In Section 2 we briefly describe graphon-estimation methods and other related work. Next, in Section 3 we formally describe our setup and introduce our algorithms. Section 4.1 contains some theory for these algorithms. In Section 5 we provide simulations and real data examples. We conclude with a discussion in Section 6.

## 2   Related work

The focus of this paper is on 1) clustering networks which have node correspondence based on estimating the underlying graphon and 2) clustering networks without node correspondence based on global properties of the networks. In this section we first cite two methods of obtaining graphon estimates, which we will use in our first algorithm. Second, we cite existing work that summarizes a network using different statistics and compares those to obtain a measure of similarity.

A prominent estimator of graphons is the so called Universal Singular Value Thresholding (USVT) estimator proposed by [9]. The main idea behind USVT is to essentially estimate the low rank structure of the population matrix by thresholding the singular values of the observed matrix at an universal cutoff, and then use retained singular values and the corresponding singular vectors to construct an estimate of the population matrix.

Another recent work [29] proposes a novel, statistically consistent and computationally efficient approach for estimating the link probability matrix by neighborhood smoothing.

Typically for large networks USVT is a lot more scalable than the neighborhood-smoothing approach. There are several other methods for graphon estimation, e.g., by fitting a stochastic blockmodel [24]. These methods can also be used in our algorithm.

In [10], a graph-based method for change-point detection is proposed, where an independent sequence of observations are considered. These are generated i.i.d. under the null hypothesis, whereas under the alternative, after a change point, the underlying distribution changes. The goal is to find this change point. The observations can be high-dimensional vectors or even networks, with the latter bearing some resemblance with our first framework. This can be viewed as clustering the observations into "past" and "future". We remark here that our graphon-estimation based clustering algorithm suggests an alternative method for change point detection in networks, namely by looking at the second eigenvector of the distance matrix between estimated graphons. Another related work is due to [14] which aims to extend the classical large sample theory to model network-valued objects.

For comparing global properties of networks, there have been many interesting works that featurize networks, see, for instance, [3]. In the Computer Science literature, graph kernels have gained much attention [27, 26]. In these works the goal is to efficiently compute different types of kernel based similarity measures (exact or approximate) between networks.

## 3 A framework for clustering networks

Let $G$ be a binary random network or graph with $n$ nodes. Denote by $A$ its adjacency matrix, which is an $n$ by $n$ symmetric matrix with binary entries. That is, $A_{ij} = A_{ji} \in \{0,1\}, 1 \leq i < j \leq n$, where $A_{ij} = 1$ if there is an observed edge between nodes $i$ and $j$, and $A_{ij} = 0$ otherwise. All the diagonal elements of $A$ are structured to be zero (i.e. $A_{ii} = 0$). We assume the following random Bernoulli model with

$$A_{ij} \mid P_{ij} \sim \text{Bernoulli}(P_{ij}), \ i < j, \tag{1}$$

where $P_{ij} = P(A_{ij} = 1)$ is the probability of link formation between nodes $i$ and $j$. We denote the link probability matrix as $P = ((P_{ij}))$. The edge probabilities are often modeled using the so-called *graphons*. A graphon $f$ is a nonnegative bounded, measurable symmetric function $f : [0,1]^2 \to [0,1]$. Given such an $f$, one can use the model

$$P_{ij} = f(\xi_i, \xi_j), \tag{2}$$

where $\xi_i, \xi_j$ are *i.i.d. uniform random variables* on $(0,1)$. In fact, any (infinite) exchangeable network arises in this way (by Aldous-Hoover representation [2, 16]).

Our current work focuses on the problem of *clustering networks*. Unlike in a traditional setup, where one observes a single network (with potentially growing number of nodes) and the goal often is to cluster the nodes, here we observe multiple networks and are interested in clustering these networks viewed as fundamental data units.

### 3.1 Node correspondence present

A simple and natural model for this is what we call the *graphon mixture model* for obvious reasons: there are only $K$ (fixed) underlying graphons $f_1, \ldots, f_K$ giving rise to link probability matrices $\Pi_1, \ldots, \Pi_K$ and we observe $T$ networks sampled i.i.d. from the mixture model

$$\mathbb{P}_{mix}(A) = \sum_{i=1}^{K} q_i \mathbb{P}_{\Pi_i}(A), \tag{3}$$

where the $q_i$'s are the mixing proportions and $\mathbb{P}_P(A) = \prod_{u<v} P_{uv}^{A_{uv}}(1 - P_{uv})^{1-A_{uv}}$ is the probability of observing the adjacency matrix $A$ when the link probability matrix is given by $P$. Consider $n$ nodes, and $T$ independent networks $A_i, i \in [T]$, which define edges between these $n$ nodes. We propose the following simple and general algorithm (Algorithm 1) for clustering them.

---

**Algorithm 1** Network Clustering based on Graphon Estimates (NCGE)

---

1: **Graphon estimation.** Given $A_1, \ldots, A_T$, estimate their corresponding link probability matrices $P_1, \ldots, P_T$ using any one of the 'blackbox' algorithms such as USVT ([9]), the neighborhood smoothing approach by [29] etc. Call these estimates $\hat{P}_1, \ldots, \hat{P}_T$.

2: **Forming a distance matrix.** Compute the $T$ by $T$ distance matrix $\hat{D}$ with $\hat{D}_{ij} = \|\hat{P}_i - \hat{P}_j\|_F$, where $\|\cdot\|_F$ is the Frobenius norm.

3: **Clustering.** Apply the spectral clustering algorithm to the distance matrix $\hat{D}$.

---

We will from now on denote the above algorithm with the different graphon estimation ('blackbox') approaches as follows: the algorithm with USVT as blackbox will be denoted by CL-USVT and the one with the neighborhood smoothing method as blackbox will be denoted by CL-NBS. We will compare these two algorithms with the CL-NAIVE method which does not estimate the underlying graphon, but clusters vectorized adjacency matrices directly (in the spirit of [10]).

### 3.2 Node correspondence absent

We use certain graph statistics to construct a feature vector. The basic statistics we choose are the trace of powers of the adjacency matrix, suitably normalized and we call them *graph moments*:

$$m_k(A) = \text{trace}(A/n)^k. \tag{4}$$

These statistics are related to various path/subgraph counts. For example, $m_2(A)$ is the normalized count of the total number of edges, $m_3(A)$ is the normalized triangle count of $A$. Higher order moments are actually counts of closed walks (or directed circuits).

The reason we use graph moments instead of subgraph counts is that the latter are quite difficult to compute and present day algorithms work only for subgraphs up to size 5. On the contrary, graph moments are easy to compute as they only involve matrix multiplication.

While it may seem that this is essentially the same as comparing the eigenspectrum, it is not clear how many eigenvalues one should use. Even if one could estimate the number of large eigenvalues using an USVT type estimator, the length is different for different networks. The trace takes into account the relative magnitudes of the eigenvalues naturally. In fact, we tried (see Section 5) using the top few eigenvalues as the sole features, but the results were not as satisfactory as using $m_k$.

We now present our second algorithm (Algorithm 2). In step 2 below we take $d$ to be the standard Euclidean metric.

---

**Algorithm 2** Network Clustering based on Log Moments (NCLM)

---

1: **Moment calculation.** For each network $A_i, i \in [T]$ and a positive integer $J$, compute the feature vector $g_J(A_i) := (\log m_1(A_i), \log m_2(A_i), \ldots, \log m_J(A_i))$.

2: **Forming a distance matrix.** For some metric $d$, set $\hat{D}_{ij} = d(g_J(A_i), g_J(A_j))$.

3: **Clustering.** Apply the spectral clustering algorithm to the distance matrix $\hat{D}$.

---

**Note:** The rationale behind taking a logarithm of the graph moments is that if we have two graphs with the same degree density but different sizes, then the degree density will not play any role in the the distance (which is necessary because the degree density will subdue any other differences otherwise). The parameter $J$ counts, in some sense, the effective number of eigenvalues we are using.

## 4 Theory

We will only mention our main results and discuss some of the consequences here. All the proofs and further details can be found in the supplementary article [1].

## 4.1 Results on NCGE

We can think of $\hat{D}_{ij}$ as estimating $D_{ij} = \|P_i - P_j\|_F$.

**Theorem 4.1.** *Suppose $D = ((D_{ij}))$ has rank $K$. Let $V$ (resp. $\hat{V}$) be the $T \times K$ matrix whose columns correspond to the leading $K$ eigenvectors (corresponding to the $K$ largest-in-magnitude eigenvalues) of $D$ (resp. $\hat{D}$). Let $\gamma = \gamma(K, n, T)$ be the $K$-th smallest eigenvalue value of $D$ in magnitude. Then there exists an orthogonal matrix $\hat{O}$ such that*

$$\|\hat{V}\hat{O} - V\|_F^2 \leq \frac{64T}{\gamma^2} \sum_i \|\hat{P}_i - P_i\|_F^2.$$

**Corollary 4.2.** *Assume for some absolute constants $\alpha, \beta > 0$ the following holds for each $i \in [T]$:*

$$\frac{\|\hat{P}_i - P_i\|_F^2}{n^2} \leq C_i n^{-\alpha} (\log n)^\beta, \tag{5}$$

*either in expectation or with high probability ($\geq 1 - \epsilon_{i,n}$). Then in expectation or with high probability ($\geq 1 - \sum_i \epsilon_{i,n}$) we have that*

$$\|\hat{V}\hat{O} - V\|_F^2 \leq \frac{64 C_T T^2 n^{2-\alpha} (\log n)^\beta}{\gamma^2}, \tag{6}$$

*where $C_T = \max_{i \leq i \leq T} C_i$.*

If there are $K$ (fixed, not growing with $T$) underlying graphons, then the constant $C_T$ does not depend on $T$. Table 1 reports values of $\alpha$, $\beta$ for various graphon estimation procedures (under assumptions on the underlying graphons, that are described in the supplementary article [1]).

Table 1: Values of $\alpha$, $\beta$ for various graphon estimation procedures.

| Procedure | USVT | NBS | Minimax rate |
|-----------|------|-----|--------------|
| $\alpha$ | 1/3 | 1/2 | 1 |
| $\beta$ | 0 | 1/2 | 1 |

While it is hard to obtain an explicit lower bound on $\gamma$ in general, let us consider a simple equal weight mixture of two graphons to illustrate the relationship between $\gamma$ and separation between graphons. Let the distance between the population graphons be $dn$. Then we have $D = Z \begin{pmatrix} 0 & dn \\ dn & 0 \end{pmatrix} Z^T$, where the $i$-th row of the binary matrix $Z$ has a single one at position $l$ if network $A_i$ is sampled from $\Pi_l$. The nonzero eigenvalues of this matrix are $Tnd/2$ and $-Tnd/2$. Thus in this case $\gamma = Tnd/2$. As a result (6) becomes

$$\|\hat{V}\hat{O} - V\|_F^2 \leq \frac{256 C_T n^{-\alpha} (\log n)^\beta}{d^2}. \tag{7}$$

Let us look at a more specific case of blockmodels with the same number ($= m$) of clusters of equal sizes ($= n/m$) to gain some insight into $d$. Let $C$ be a $n \times m$ binary matrix of memberships such that $C_{ib} = 1$ if node $i$ within a blockmodel comes from cluster $b$. Consider two blockmodels $\Pi_1 = CB_1 C^T$ with $B_1 = (p - q)I_m + qE_m$ and $\Pi_2 = CB_2 C^T$ with $B_2 = (p' - q')I_m + q'E_m$, where $I_m$ is the identity matrix of order $m$ (here the only difference between the models come from link formation probabilities within/between blocks, the blocks remaining the same). In this case

$$d^2 = \frac{\|\Pi_1 - \Pi_2\|_F^2}{n^2} = \frac{1}{m}(p - p')^2 + \left(1 - \frac{1}{m}\right)(q - q')^2.$$

The bound (6) can be turned into a bound on the proportion of "misclustered" networks, defined appropriately. There are several ways to define misclustered nodes in the context of community detection in stochastic blockmodels that are easy to analyze with spectral clustering (see, e.g., [25, 18]). These definitions work in our context too. For example, if we

use Definition 4 of [25] and denote by $\mathcal{M}$ the set of misclustered networks, then from the proof of their Theorem 1, we have

$$|\mathcal{M}| \leq 8m_T \|\hat{V}\hat{O} - V\|_F^2,$$

where $m_T = \max_{j=1,\ldots,K}(Z^T Z)_{jj}$ is the maximum number of networks coming from any of the graphons.

## 4.2 Results on NCLM

We first establish concentration of $\text{trace}(A^k)$. The proof uses Talagrand's concentration inequality, which requires additional results on Lipschitz continuity and convexity. This is obtained via decomposing $A \mapsto \text{trace}(A^k)$ into a linear combination of convex-Lipschitz functions.

**Theorem 4.3** (Concentration of moments)**.** *Let $A$ be the adjacency matrix of a random graph with link-probability matrix $P$. Then for any $k$. Let $\psi_k(A) := \frac{n}{k\sqrt{2}}m_k(A)$. Then*

$$\mathbb{P}(|\psi_k(A) - \mathbb{E}\psi_k(A)| > t) \leq 4\exp(-(t - 4\sqrt{2})^2/16).$$

As a consequence of this, we can show that $g_J(A)$ concentrates around $\bar{g}_J(A) := (\log \mathbb{E}m_2(A), \ldots, \log \mathbb{E}m_J(A))$.

**Theorem 4.4** (Concentration of $g_J(A)$)**.** *Let $\mathbb{E}A = \rho S$, where $\rho \in (0,1)$, $\min_{i,j} S_{ij} = \Omega(1)$, and $\sum_{i,j} S_{ij} = n^2$. Then $\|\bar{g}_J(A)\| = \Theta(J^{3/2}\log(1/\rho))$, and for any $0 < \delta < 1$ satisfying $\delta J \log(1/\rho) = \Omega(1)$, we have*

$$\mathbb{P}(\|g_J(A) - \bar{g}_J(A)\| \geq \delta J^{3/2}\log(1/\rho)) \leq JC_1 e^{-C_2 n^2 \rho^{2J}}.$$

We expect that $\bar{g}_J$ will be a good population level summary for many models. In general, it is hard to show an explicit separation result for $\bar{g}_J$. However, in simple models, we can do explicit computations to show separation. For example, in a two parameter blockmodel $B = (p-q)I_m + qE_m$, with equal block sizes, we have $\mathbb{E}m_2(A) = (p/m + (m-1)q/m)(1+o(1))$, $\mathbb{E}m_3(A) = (p^3/m^2 + (m-1)pq^2/m^2 + (m-1)(m-2)q^3/6m^2)(1 + o(1))$ and so on. Thus we see that if $m = 2$, then $\bar{g}_2$ should be able to distinguish between such blockmodels (i.e. different $p$, $q$).

**Note:** After this paper was submitted, we came to know of a concurrent work [20] that provides a topological/combinatorial perspective on the expected graph moments $\mathbb{E}m_k(A)$. Theorem 1 in [20] shows that under some mild assumptions on the model (satisfied, for example, by generalized random graphs with bounded kernels as long as the average degree grows to infinity), $\mathbb{E}\text{trace}(A^k) = \mathbb{E}(\# \text{ of closed } k\text{-walks})$ will be asymptotic to $\mathbb{E}(\# \text{ of closed } k\text{-walks that trace out a } k\text{-cycle}) plus \mathbf{1}_{\{k \text{ even}\}}\mathbb{E}(\# \text{ of closed } k\text{-walks that trace out a } (k/2+1)\text{-tree})$. For even $k$, if the degree grows fast enough, $k$-cycles tend to dominate, whereas for sparser graphs trees tend to dominate. From this and our concentration results, we can expect NCLM to be able to tell apart graphs which are different in terms the counts of these simpler closed $k$-walks. Incidentally, the authors of [20] also show that the expected count of closed non-backtracking walks of length $k$ is dominated by walks tracing out $k$-cycles. Thus if one uses counts of closed non-backtracking $k$-walks (i.e. moments of the non-backtracking matrix) instead of just closed $k$-walks as features, one would expect similar performance on denser networks, but in sparser settings it may lead to improvements because of the absence of the non-informative trees in lower order even moments.

## 5 Simulation study and data analysis

In this section, we describe the results of our experiments with simulated and real data to evaluate the performance of NCGE and NCLM. We measure performance in terms of clustering error which is the minimum normalized hamming distance between the estimated label vector and all $K!$ permutations of the true label assignment. Clustering accuracy is one minus clustering error.

**Node correspondence present:** We provide two simulated data experiments[1] for clustering networks with node correspondence. In each experiment twenty 150-node networks were generated from a mixture of two graphons, 13 networks from the first and the other 7 from the second. We also used a scalar multiplier with the graphons to ensure that the networks are not too dense. The average degree for all these experiments were around 20-25. We report the average error bars from a few random runs.

First we generate a mixture of graphons from two blockmodels, with probability matrices $(p_i - q_i)I_m + q_i E_m$ with $i \in \{1, 2\}$. We use $p_2 = p_1(1 + \epsilon)$ and $q_2 = q_1(1 + \epsilon)$ and measure clustering accuracy as the multiplicative error $\epsilon$ is increased from 0.05 to 0.15. We compare CL-USVT, CL-NBS and CL-NAIVE and the results are summarized in Figure 1(A). We have observed two things. First, CL-USVT and CL-NBS start distinguishing the graphons better as $\epsilon$ increases (as the theory suggests). Second, the naive approach does not do a good job even when $\epsilon$ increases.

Figure 1: We show the behavior of the three algorithms when $\epsilon$ increases, when the underlying network is generated from (A) a blockmodel, and (B) a smooth graphon.

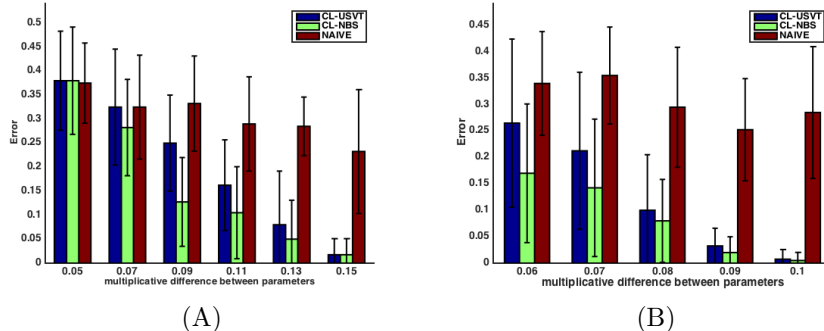

(A)                                    (B)

In the second simulation, we generate the networks from two smooth graphons $\Pi_1$ and $\Pi_2$, where $\Pi_2 = \Pi_1(1 + \epsilon)$ (here $\Pi_1$ corresponds to the graphon 3 appearing in Table 1 of [29]). As is seen from Figure 1(B), here also CL-USVT and CL-NBS outperform the naive algorithm by a huge margin. Also, CL-NBS is consistently better than CL-USVT, which shows that the accuracy of the graphon estimation procedure is important (for example, USVT is known to perform worse as the network becomes sparser).

**Node correspondence absent:** We show the efficacy of our approach via two sets of experiments. We compare our log-moment based method NCLM with three other methods. The first is Graphlet Kernels [26] with 3, 4 and 5 graphlets, denoted by GK3, GK4 and GK5 respectively. In the second method, we use six different network-based statistics to summarize each graph; these statistics are the algebraic connectivity, the local and global clustering coefficients [23], the distance distribution [19] for 3 hops, the Pearson correlation coefficient [22] and the rich-club metric [30]. We also compare against graphs summarized by the top $J$ eigenvalues of $A/n$ (TopEig). These are detailed in the supplementary article [1].

For each distance matrix $\hat{D}$ we compute with NCLM, GraphStats and TopEig, we calculate a similarity matrix $\mathcal{K} = \exp(-t\hat{D})$ where $t$ is chosen as the value, within a range, which maximizes the relative eigengap $(\lambda_K(\mathcal{K}) - \lambda_{K+1}(\mathcal{K}))/\lambda_{K+1}(\mathcal{K})$. It would be interesting to have a data dependent range for $t$.

For each matrix $\mathcal{K}$ we calculate the top few eigenvectors, say $N$ many, and do $K$-means on them to get the final clustering. We use $N = K$; however, for GK3, GK4, and GK5, we had to use a smaller $N$ which boosted their clustering accuracy.

First we construct four sets of parameters for the two parameter blockmodel (also known as the planted partition model): $\Theta_1 = \{p = 0.1, q = 0.05, K = 2, \rho = 0.6\}$, $\Theta_2 = \{p = 0.1, q =$

$0.05, K = 2, \rho = 1\}$, $\Theta_3 = \{p = 0.1, q = 0.05, K = 8, \rho = 0.6\}$, and $\Theta_4 = \{p = 0.2, q = 0.1, K = 8, \rho = 0.6\}$. Note that the first two settings differ only in the density parameter $\rho$. The second two settings differ in the within and across cluster probabilities. The first two and second two differ in $K$. For each parameter setting we generate two sets of 20 graphs, one with $n = 500$ and the other with $n = 1000$.

For choosing $J$, we calculate the moments for a large $J$; compute a kernel similarity matrix for each choice of $J$ and report the one with largest relative eigengap between the $K^{th}$ and $(K + 1)^{th}$ eigenvalue. We show these plots in the supplementary article [1]. We see that the eigengap increases and levels off after a point. However, as $J$ increases, the computation time increases, so there is a trade-off. We report the accuracy of $J = 5$, whereas $J = 8$ also returns the same in 48 seconds.

Table 2: Error of 6 different methods on the simulated networks.

|          | NCLM ($J = 5$) | GK3 | GK4  | GK5  | GraphStats ($J = 6$) | TopEig ($J = 5$) |
|----------|----------------|-----|------|------|----------------------|------------------|
| Error    | **0**          | 0.5 | 0.36 | 0.26 | 0.37                 | 0.18             |
| Time (s) | 25             | 14  | 16   | 38   | 94                   | 8                |

We see that NCLM performs the best. For GK3, GK4 and GK5, if one uses the top two eigenvectors, and clusters those into 4 groups (since there are four parameter settings), the errors are respectively 0.08, 0.025 and 0.03. This means that, for clustering, one needs to estimate the effective rank of the graphlet kernels as well. TopEig performs better than GraphStats, which has trouble separating out $\Theta_2$ and $\Theta_4$.

**Note:** Intuitively one would expect that, if there is node correspondence between the graphs, clustering based on graphon estimates would work better, because it aims to estimate the underlying probabilistic model for comparison. However, in our experiments we found that a properly tuned NCLM matched the performance of NCGE. This is probably because a properly tuned NCLM captures the global features that distinguish two graphons. We leave it for future work to compare their performance theoretically.

**Real Networks:** We cluster about fifty real world networks. We use 11 co-authorship networks between 15,000 researchers from the High Energy Physics corpus of the arXiv, 11 co-authorship networks with 21,000 nodes from Citeseer (which had Machine Learning in their abstracts), 17 co-authorship networks (each with about 3000 nodes) from the NIPS conference and finally 10 Facebook ego networks[2]. The average degrees vary between 0.2 to 0.4 for co-authorship networks and are around 10 for the ego networks. Each co-authorship network is dynamic, i.e. a node corresponds to an author in that corpus and this node index is preserved in the different networks over time. The ego networks are different in that sense, since each network is the subgraph of Facebook induced by the neighbors of a given central or "ego" node. The sizes of these networks vary between 350 to 4000.

Table 3: Clustering error of 6 different methods on a collection of real world networks consisting of co-authorship networks from Citeseer, High Energy Physics (HEP-Th) corpus of arXiv, NIPS and ego networks from Facebook.

|          | NCLM ($J = 8$) | GK3 | GK4 | GK5 | GraphStats ($J = 8$) | TopEig ($J = 30$) |
|----------|----------------|-----|-----|-----|----------------------|-------------------|
| Error    | 0.1            | 0.6 | 0.6 | 0.6 | 0.16                 | 0.32              |
| Time (s) | 2.7            | 45  | 50  | 60  | 765                  | 14                |

Table 3 summarizes the performance of different algorithms and their running time to compute distance between the graphs. We use the different sources of networks as labels, i.e. HEP-Th will be one cluster, etc. We explore different choices of $J$, and see that the best performance is from NCLM, with $J = 8$, followed closely by GraphStats. TopEig ($J$ in this case is where the eigenspectra of the larger networks have a knee) and the graph kernels do not perform very well. GraphStats take 765 seconds to complete, whereas NCLM finishes in 2.7 seconds. This is because the networks are large but extremely sparse, and so calculation of matrix powers is comparatively cheap.

Figure 2: Kernel matrix for NCLM on 49 real networks.

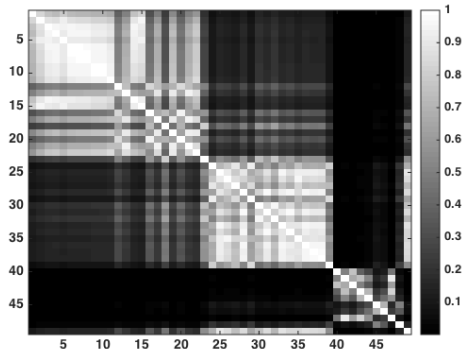

In Figure 2, we plot the kernel similarity matrix obtained using NCLM on the real networks (higher the value, more similar the points are). The first 11 networks are from HEP-Th, whereas the next 11 are from Citeseer. The next 16 are from NIPS and the remaining ones are the ego networks from Facebook. First note that {HEP-Th, Citeseer}, NIPS and Facebook are well separated. However, HEP-Th and Citeseer are hard to separate out. This is also verified by the bad performance of TopEig in separating out the first two (shown in Section 5). However, in Figure 2, we can see that the Citeseer networks are different from HEP-Th in the sense that they are not as strongly connected inside as HEP-Th.

## 6 Discussion

We consider the problem of clustering network-valued data for two settings, both of which are prevalent in practice. In the first setting, different network objects have node correspondence. This includes clustering brain networks obtained from FMRI data where each node corresponds to a specific region in the brain, or co-authorship networks between a set of authors where the connections vary from one year to another. In the second setting, node correspondence is not present, e.g., when one wishes to compare different types of networks: co-authorship networks, Facebook ego networks, etc. One may be interested in seeing if co-authorship networks are more "similar" to each other than ego or friendship networks.

We present two algorithms for these two settings based on a simple general theme: summarize a network into a possibly high dimensional feature vector and then cluster these feature vectors. In the first setting, we propose NCGE, where each network is represented using its graphon-estimate. We can use a variety of graphon estimation algorithms for this purpose. We show that if the graphon estimation is consistent, then NCGE can cluster networks generated from a finite mixture of graphons in a consistent way, if those graphons are sufficiently different. In the second setting, we propose to represent a network using an easy-to-compute summary statistic, namely the vector of the log-traces of the first few powers of a suitably normalized version of the adjacency matrix. We call this method NCLM and show that the summary statistic concentrates around its expectation, and argue that this expectation should be able to separate networks generated from different models. Using simulated and real data experiments we show that NCGE is vastly superior to the naive but often-used method of comparing adjacency matrices directly, and NCLM outperforms most computationally expensive alternatives for differentiating networks without node correspondence. In conclusion, we believe that these methods will provide practitioners with a powerful and computationally tractable tool for comparing network-structured data in a range of disciplines.

**Acknowledgments**

We thank Professor Peter J. Bickel for helpful discussions. SSM was partially supported by NSF-FRG grant DMS-1160319 and a Loéve Fellowship. PS was partially supported by NSF grant DMS 1713082. LL was partially supported by NSF grants IIS 1663870, DMS 1654579 and a DARPA grant N-66001-17-1-4041.

## Footnotes

[1]Code used in this paper is publicly available at `https://github.com/soumendu041/clustering-network-valued-data`.

[2]https://snap.stanford.edu/data/egonets-Facebook.html

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
