[Supplementary Material · nips-2017-supplementary-file.pdf]

# Supplement to "On clustering network-valued data"

**Soumendu Sundar Mukherjee**
Department of Statistics
University of California, Berkeley
Berkeley, California 94720, USA
soumendu@berkeley.edu

**Purnamrita Sarkar**
Department of Statistics and Data Sciences
University of Texas, Austin
Austin, Texas 78712, USA
purna.sarkar@austin.utexas.edu

**Lizhen Lin**
Department of Applied and Computational Mathematics and Statistics
Univeristy of Notre Dame
Notre Dame, Indiana 46556, USA
lizhen.lin@nd.edu

## Abstract

This supplementary article contains proofs of the main results of our paper "On clustering network-valued data". Some further details on our experimental results are also provided.

## A   Proofs and related discussions

### A.1   NCGE

**Proposition A.1.** *We have*

$$\|\hat{D} - D\|_F^2 \le 4T \sum_i \|\hat{P}_i - P_i\|_F^2.$$

*Proof.* The proof is straightforward. By triangle inequality we have

$$|\hat{D}_{ij} - D_{ij}| = \left| \|\hat{P}_i - \hat{P}_j\|_F - \|P_i - P_j\|_F \right| \le \|\hat{P}_i - P_i\|_F + \|\hat{P}_j - P_j\|_F.$$

Therefore

$$\|\hat{D} - D\|_F^2 = \sum_{i,j} |\hat{D}_{ij} - D_{ij}|^2 \le \sum_{i,j} (\|\hat{P}_i - P_i\|_F + \|\hat{P}_j - P_j\|_F)^2$$
$$\le 2 \sum_{i,j} (\|\hat{P}_i - P_i\|_F^2 + \|\hat{P}_j - P_j\|_F^2) = 4T \sum_i \|\hat{P}_i - P_i\|_F^2.$$

$\square$

**Proposition A.2** (Davis-Kahan). *Suppose $D$ has rank $K$. Let $V$ (resp. $\hat{V}$) be the $T \times K$ matrix whose columns correspond to the leading $K$ eigenvectors (corresponding to the $K$ largest-in-magnitude eigenvalues) of $D$ (resp. $\hat{D}$). Let $\gamma = \gamma(K, n, T)$ be the $K$-th smallest eigenvalue value of $D$ in magnitude. Then there exists an orthogonal matrix $\hat{O}$ such that*

$$\|\hat{V}\hat{O} - V\|_F \le \frac{4\|\hat{D} - D\|_F}{\gamma}.$$

*Proof.* This follows from a slight variant of Davis-Kahan theorem that appears in [5]. Since $D$ is a Euclidean distance matrix of rank $K$, its eigenvalues must be of the form

$$\lambda_1 \geq \cdots \geq \lambda_u > 0 = \cdots = 0 > \lambda_v \geq \cdots \geq \lambda_n,$$

with $u + n - v + 1 = K$. Applying Theorem 2 of [5] with $r = 1$, $s = u$ we get that if $V_+$ denotes matrix whose columns are the eigenvectors of $D$ corresponding to $\lambda_1, \ldots, \lambda_u$ and $\hat{V}_+$ denotes the corresponding matrix for $\hat{D}$, then there exists an orthogonal matrix $\hat{O}_+$ such that

$$\|\hat{V}_+\hat{O}_+ - V_+\|_F \leq \frac{2\sqrt{2}\|\hat{D} - D\|_F}{\lambda_u}.$$

Similarly, considering the eigenvalues $\lambda_v, \ldots, \lambda_n$, and applying Theorem 2 of [5] with $r = v$ and $s = n$ we get that

$$\|\hat{V}_-\hat{O}_- - V_-\|_F \leq \frac{2\sqrt{2}\|\hat{D} - D\|_F}{-\lambda_v},$$

where $V_-$, $\hat{V}_-$ and $\hat{O}_-$ are the relevant matrices. Set $V = [V_+ : V_-]$, $\hat{V} = [\hat{V}_+ : \hat{V}_-]$ and $\hat{O} = \begin{pmatrix} \hat{O}_+ & 0 \\ 0 & \hat{O}_- \end{pmatrix}$. Then note that the columns of $V$ are eigenvectors of $D$ corresponding to its $K$ largest-in-magnitude eigenvalues, that $O$ is orthogonal and also that $\gamma = \min\{\lambda_u, -\lambda_v\}$. Thus

$$\begin{aligned} \|\hat{V}\hat{O} - V\|_F^2 &= \|\hat{V}_+\hat{O}_+ - V_+\|_F^2 + \|\hat{V}_-\hat{O}_- - V_-\|_F^2 \\ &\leq \frac{8\|\hat{D} - D\|_F^2}{\lambda_u^2} + \frac{8\|\hat{D} - D\|_F^2}{\lambda_v^2} \\ &\leq \frac{16\|\hat{D} - D\|_F^2}{\gamma^2}, \end{aligned}$$

which is the desired bound. $\qquad\square$

*Proof of Theorem 4.1.* Follows immediately from Propositions A.1-A.2. $\qquad\square$

**Proposition A.3.** *Suppose there are $K$ underlying graphons, as in the graphon mixture model (3), and assume that in our sample there is at least one representative from each of these. Then $D$ has the form $Z\mathcal{D}Z^T$ where the ith row of the binary matrix $Z$ has a single one at position $l$ if network $A_i$ is sampled from $\Pi_l$, and $\mathcal{D}$ is the $K \times K$ matrix of distances between the $\Pi_l$. As a consequence $D$ is of rank $K$. Then there exists a $T \times K$ matrix $V$ whose columns are eigenvectors of $D$ corresponding to the $K$ nonzero eigenvalues, such that*

$$V_{i\star} = V_{j\star} \Leftrightarrow Z_{i\star} = Z_{j\star},$$

*so that knowing $V$, one can recover the clusters perfectly.*

*Proof.* The proof is standard. Note that $Z^T Z$ is positive definite. Consider the matrix $(Z^T Z)^{1/2} B (Z^T Z)^{1/2}$ and let $U\Delta U^T$ be its spectral decomposition. Then the matrix $V = Z(Z^T Z)^{-1/2}U$ has the required properties. $\qquad\square$

**USVT:** Theorem 2.7 of [3] tells us that if the underlying graphons are Lipschitz then $\mathbb{E}\frac{\|\hat{P}_i - P_i\|_F^2}{n^2} \leq C_i n^{-1/3}$, where the constant $C_i$ depends only on the Lipschitz constant of the underlying graphon $f_i$. So, the condition (5) of Corollary 4.2 is satisfied with $\alpha = 1/3$, $\beta = 0$.

**Neighborhood smoothing:** The authors of [6] work with a class $\mathcal{F}_{\delta,L}$ of piecewise Lipschitz graphons, see Definition 2.1 of their paper. The proof of Theorem 2.2 of [6] reveals that if $f_i \in \mathcal{F}_{\delta_i, L_i}$, then there exist a global constant $C$ and constants $C_i \equiv C_i(L_i)$, such that for all $n \geq N_i \equiv N_i(\delta_i)$, with probability at least $1 - n^{-C}$, we have $\frac{\|\hat{P}_i - P_i\|_F^2}{n^2} \leq C_i\sqrt{\frac{\log n}{n}}$. Thus the condition (5) of Corollary 4.2 is satisfied with $\alpha = \beta = 1/2$, for all $n \geq N_T := \max_{1 \leq i \leq T} N_i$.

**Remark A.1.** *In the case of the graphon mixture model (3), the constants $C_T$ and $N_T$ will be free of $T$ as there are only $K$ (fixed) underlying graphons. Also, if each $f_i \in \mathcal{F}_{\delta, L_i}$, then $N_T$ will not depend on $T$ and if the Lipschitz constants are the same for each graphon, then $C_T$ will not depend on $T$.*

**Remark A.2.** *There are combinatorial algorithms that can achieve the minimax rate, $n^{-1} \log n$, of graphon estimation [4]. Albeit impractical, these algorithms can be used to achieve the optimal bound of $4C_T n^{-1} \log n$ in Proposition A.1.*

**Remark A.3.** *We do not expect CL-NAIVE to perform well simply because $A$ is not a good estimate of $P$ in Frobenius norm in general. Indeed,*

$$\frac{1}{n^2} \mathbb{E} \|A - P\|_F^2 = \frac{1}{n^2} \sum_{i,j} \mathbb{E}(A_{ij} - P_{ij})^2 = \frac{1}{n^2} \sum_{i \neq j} P_{ij}(1 - P_{ij}) + \frac{1}{n^2} \sum_i P_{ii}^2 \leq \frac{1}{4}(1 + o(1)),$$

*with equality, for example, when each $P_{ij} = \frac{1}{2} + o(1)$.*

## A.2  NCLM

Note that $\text{trace}(A^k)$ counts the number of closed $k$-walks (or directed circuits) in the graph corresponding to $A$. Figure 1 shows the circuits corresponding to $k = 4$.

Figure 1: Circuits related to $m_4(A)$.

(A)            (B)            (C)            (D)

**Proposition A.4** (Lipschitz continuity). *Suppose $A$ and $A + U$ are matrices with entries in $[-1, 1]$, then*

1. *$|\text{trace}((A + U)_+^k) - \text{trace}(A_+^k)| \leq kn^{k-1}\|U\|_F$,*

2. *$|\text{trace}((A + U)_-^k) - \text{trace}(A_-^k)| \leq kn^{k-1}\|U\|_F$,*

*i.e. the map $\phi_{+,k} : S_{[-1,1]}^{n \times n} \to [0, \infty)$ defined on the space $S_{[-1,1]}^{n \times n}$ of symmetric matrices with entries in $[-1, 1]$ by $\phi_{+,k}(A) = \text{trace}(A_+^k)$ is Lipschitz with constant $kn^{k-1}$. Note that this implies that the map $\tilde{\phi}_{+,k} : [0,1]^{n(n-1)/2} \to [0, \infty)$ defined by*

$$\tilde{\phi}_{+,k}((a_{ij})_{1 \leq i < j \leq n}) = \text{trace}(A_+^k),$$

*where $A_{ij} = A_{ji} = a_{ij}$, for $1 \leq i < j \leq n$ and $A_{ii} = 0$, is Lipschitz with constant $\sqrt{2}kn^{k-1}$. The same statements hold for analogously defined $\phi_{-,k}$ and $\tilde{\phi}_{-,k}$.*

*Proof.* Let $\lambda_1 \geq \cdots \geq \lambda_n$ be the ordered eigenvalues of $A + U$, whereas $\nu_1 \geq \cdots \geq \nu_n$ be the ordered eigenvalues of $A$. Let $\sigma_1 \geq \cdots \geq \sigma_n$ be the ordered eigenvalues of $U$. First note that since these matrices have entries in $[-1, 1]$, their Frobenius norm is at most $n$. Thus all their eigenvalues are in $[-n, n]$.

We now compute the derivative of $\text{trace}A^k$ with respect to a particular variable $A_{ij}$. We claim that

$$\frac{d}{dA_{ij}} \text{trace}A^k = 2kA_{ij}^{k-1}.$$

To do this we shall work with the linear map interpretation of derivative (in this case multiplication by a number). First consider the map $f : \mathbb{R}^{n \times n} \to \mathbb{R}^{n \times n}$ defined as $f(A) = A^k$.

Then consider the map $g : \mathbb{R}^{n \times n} \to \mathbb{R}$ defined by $g(A) = \text{trace}(A)$. Finally consider the map $h : \mathbb{R} \to \mathbb{R}^{n \times n}$ defined as $h(x) = A + (x - A_{ij})e_i e_j^T + (x - A_{ij})e_j e_i^T$. Now we view the function $\text{trace} A^k$ for $A$ symmetric as a function of $A_{ij}$ as $g \circ f \circ h(A_{ij})$. Therefore, by chain rule

$$\frac{d}{dA_{ij}} \text{trace} A^k(u) = D_{f \circ h(A_{ij})} g \circ D_{h(A_{ij})} f \circ D_{A_{ij}} h(u).$$

Now $g$ is a linear function. Therefore $D_A g = g$. On the other hand, it is easy to see that $D_{A_{ij}} h(u) = u e_i e_j^T + u e_j e_i^T$. Finally $f$ can be viewed as the composition of two maps $\alpha(A_1, \ldots, A_k) = A_1 A_2 \cdots A_k$ and $\beta(A) = (A, \ldots, A)$. Notice that $D_{(A_1, \ldots, A_k)} \alpha(H_1, \ldots, H_k) = H_1 A_2 \cdots A_k + A_1 H_2 \cdots A_k + \ldots + A_1 A_2 \cdots H_k)$, and $D_A \beta = \beta$. Thus

$$D_A f(H) = D_{\beta(A)} \alpha \circ D_A \beta(H) = D_{\beta(A)} \alpha(H, \ldots, H) = H A^{k-1} + \cdots + H A^{k-1} = k H A^{k-1}.$$

Therefore

$$\frac{d}{dA_{ij}} \text{trace} A^k(u) = g \circ D_{h(A_{ij})} f(u e_i e_j^T + u e_j e_i^T).$$

Noting that $h(A_{ij}) = A$ we get

$$\frac{d}{dA_{ij}} \text{trace} A^k(u) = g(k(u e_i e_j^T + u e_j e_i^T) A^{k-1}) = 2k A_{ij}^{k-1} u.$$

Therefore the map $\tilde{\phi}_k : \mathbb{R}^{n(n-1)/2} \to \mathbb{R}$ defined by $\tilde{\phi}_k((A_{ij})_{i<j}) \equiv \tilde{\phi}_k(A) = \text{trace}(A^k)$ has gradient $2k(A_{ij}^{k-1})_{i<j}$.

Therefore

$$|\tilde{\phi}_k(A) - \tilde{\phi}_k(B)| \le \|\nabla \tilde{\phi}_k\|_2 \|(A_{ij}) - (B_{ij})\|_2.$$

But $\|\nabla \tilde{\phi}_k\|_2 = \sqrt{2}k\|A^{k-1}\|_F \le \sqrt{2}k n^{k-1}$ (by repeated application of the inequality $\|XY\|_F \le \|X\|_F \|Y\|_{op}$ and noting that $\|A\|_F \le n$), and $\|(A_{ij}) - (B_{ij})\|_2 = \|A - B\|_F / \sqrt{2}$. Therefore

$$|\tilde{\phi}_k(A) - \tilde{\phi}_k(B)| \le k n^{k-1} \|A - B\|_F.$$

Also as before

$$\begin{aligned}
|\text{trace}(A + U)_+^k - \text{trace} A_+^k| &= |\tilde{\phi}_k((A + U)_+) - \tilde{\phi}_k(A_+)| \\
&\le k n^{k-1} \|(A + U)_+ - A_+\|_F \\
&\le k n^{k-1} \|U\|_F,
\end{aligned}$$

where in the last step we have used the fact that $A \mapsto A_+$ is projection onto the PSD cone and hence non-expansive (i.e. 1-Lipschitz). Part 2. may now be obtained easily by noting that $A_- = (-A)_+$. $\qquad \square$

**Proposition A.5** (Convexity). *The functions $\phi_{\pm,k}$ and $\tilde{\phi}_{\pm,k}$ are convex on their respective domains.*

*Proof.* We recall the standard result that if a continuous map $t \mapsto f(t)$ is convex, so is $A \mapsto \text{trace} f(A)$ on the space of Hermitian matrices, and it is strictly convex if f is strictly convex (See, for example, Theorem 2.10 of [2]). To use this we note that $x \mapsto x_+^k$ is continuous and convex, and so is $x \mapsto x_-^k$. This establishes convexity of $\phi_{\pm,k}$. Convexity of $\tilde{\phi}_{\pm,k}$ is an immediate consequence. $\qquad \square$

*Proof of Theorem 4.3.* The idea is to use Talagrand's concentration inequality for convex-Lipschitz functions (cf. [1], Theorem 7.12). First note that for $k$ even, we have

$$\psi_k(A) = \psi_{+,k}(A) + \psi_{-,k}(A)$$

and for $k$ odd

$$\psi_k(A) = \psi_{+,k}(A) - \psi_{-,k}(A),$$

where

$$\psi_{\pm,k}(A) = \frac{1}{\sqrt{2}k n^{k-1}} \tilde{\phi}_{\pm,k}(A).$$

Viewed as a map from $[0,1]^{n(n-1)/2}$ to $[0,\infty)$, both $\psi_{\pm,k}$ are convex, 1-Lipschitz. Therefore, by Talagrand's inequality,

$$\mathbb{P}(|\psi_{\pm,k}(A) - \mathbb{M}\psi_{\pm,k}(A)| > t) \le 2\exp(-t^2/4),$$

where $\mathbb{M}\psi_{\pm,k}(A)$ is a median of $\psi_{\pm,k}(A)$. By Exercise 2.2 of [1], we have

$$|\mathbb{M}\psi_{\pm,k}(A) - \mathbb{E}\psi_{\pm,k}(A)| \le 2\sqrt{2},$$

which implies that

$$\mathbb{P}(|\psi_{\pm,k}(A) - \mathbb{E}\psi_{\pm,k}(A)| > t) \le 2\exp(-(t - 2\sqrt{2})^2/4).$$

Therefore

$$\mathbb{P}(|\psi_k(A) - \mathbb{E}\psi_k(A)| > t) \le \mathbb{P}(|\psi_{+,k}(A) - \mathbb{E}\psi_{+,k}(A)| > t/2) + \mathbb{P}(|\psi_{-,k}(A) - \mathbb{E}\psi_{-,k}(A)| > t/2)$$
$$\le 4\exp(-(t - 4\sqrt{2})^2/16),$$

as desired. $\qquad\square$

**Proposition A.6** (Order of expectation). *Let $\mathbb{E}A = P = \rho S$, where $\rho \in (0,1)$, $\min_{i,j} S_{ij} = \Omega(1)$, and $\sum_{i,j} S_{ij} = n^2$. Then,*

$$\rho^k \preceq \mathbb{E}m_k(A) \preceq \rho^{k-1}.$$

*Proof.* Note that

$$\mathrm{trace}(A^k) = \sum_{i_1, i_2, \ldots, i_k} A_{i_1 i_2} A_{i_2 i_3} \cdots A_{i_k i_1}.$$

Since $A_{ij}$'s are Bernoulli random variables, Letting $P_\star := \min_{ij} P_{ij}$ and $P_\# := \max P_{ij}$, we see that

$$P_\star^\ell \le \mathbb{E}A_{i_1 i_2} A_{i_2 i_3} \cdots A_{i_k i_1} \le P_\#^\ell,$$

where $1 \le \ell \le k$ is the number of distinct sets in among $\{i_1, i_2\}, \{i_2, 1_3\}, \ldots, \{i_k, i_1\}$. We call $\ell$ the weight of the sequence $i_1, \ldots, i_k$. We can easily see that the total number of sequences is bounded by $n^k$, and the number of sequences with weight $\ell$, call it $N(\ell; k, n)$, is bounded above by $n^{\ell+1}$. In fact,

$$N(k; k, n) = n(n-1)(n-2)^{k-3}(n-3) \asymp n^k.$$

We thus have

$$\sum_{\ell=1}^{k} N(\ell; k, n) P_\star^\ell \le \mathbb{E}\mathrm{trace}(A^k) \le \sum_{\ell=1}^{k} N(\ell; k, n) P_\#^\ell.$$

This gives us trivial upper and lower bounds ($C_1, C_2 > 0$ are absolute constants whose values are adjusted as necessary)

$$C_1 n^k \rho^k \le C_1 n^k P_\star^k \le N(k; k, n) \le \mathbb{E}\mathrm{trace}(A^k) \le \sum_{\ell=1}^{k-1} n^{\ell+1} P_\#^\ell + n^k P_\#^k$$
$$= n\frac{(nP_\#)^k - (nP_\#)}{nP_\# - 1} + n^k P_\#^k$$
$$\le C_2 n^k P_\#^{k-1} \le C_2 n^k \rho^{k-1}.$$

This completes the proof. $\qquad\square$

In the following, we will again use $C_1, C_2 > 0$ as absolute constants whose values may change from line to line.

*Proof of Theorem 4.4.* First of all, by Proposition A.6 we have
$$|\log \mathbb{E} m_k(A)| = \Theta(k \log(1/\rho)),$$
from which we conclude that $\|\bar{g}_J(A)\| = \Theta(J^{3/2} \log(1/\rho))$.

Writing $\mu_k = \mathbb{E} m_k(A)$, and using Theorem 4.3, we get

$$\mathbb{P}(|\log m_k(A) - \log \mu_k| > t) = \mathbb{P}(\frac{m_k}{\mu_k} - 1 > e^t - 1) + \mathbb{P}(\frac{m_k}{\mu_k} - 1 < -(1 - e^{-t}))$$

$$\leq \mathbb{P}(|m_k - \mu_k| > (e^t - 1)\mu_k) + \mathbb{P}(|m_k - \mu_k| > (1 - e^{-t})\mu_k)$$

$$\leq C_1 e^{-C_2 \frac{n^2 \mu_k^2 (e^t - 1)^2}{k^2}} + C_1 e^{-C_2 \frac{n^2 \mu_k^2 (1 - e^{-t})^2}{k^2}}$$

$$\leq C_1 e^{-C_2 \frac{n^2 \rho^{2k} (e^t - 1)^2}{k^2}} + C_1 e^{-C_2 \frac{n^2 \rho^{2k} (1 - e^{-t})^2}{k^2}},$$

where in the last line we have used Proposition A.6. Using this along with an union bound, we get

$$\mathbb{P}(\|g_J(A) - \bar{g}_J(A)\| \geq t) \leq \sum_{k=2}^{J} \mathbb{P}(|\log m_k(A) - \log \mathbb{E} m_k(A)| > \frac{t}{\sqrt{J}})$$

$$\leq \sum_{k=2}^{J} C_1 e^{-C_2 \frac{n^2 \rho^{2k} (e^{t/\sqrt{J}} - 1)^2}{k^2}} + C_1 e^{-C_2 \frac{n^2 \rho^{2k} (1 - e^{-t/\sqrt{J}})^2}{k^2}}.$$

Choosing $t = \delta J^{3/2} \log(1/\rho)$, where $\delta J \log(1/\rho) = \Omega(1)$, we see that $e^{t/\sqrt{J}} - 1 = \rho^{-\delta J} - 1 = \Omega(\rho^{-\delta J})$ and $1 - e^{-t/\sqrt{J}} = 1 - \rho^{\delta J} = \Omega(1)$. Also note that $\rho^{2k}/k^2 \geq \rho^{2J}/4$. Therefore we have

$$\mathbb{P}(\|g_J(A) - \bar{g}_J(A)\| \geq \delta J^{3/2} \log(1/\rho)) \leq J(C_1 e^{-C_2 n^2 \rho^{2J} \rho^{-2\delta J}} + C_1 e^{-C_2 n^2 \rho^{2J}})$$

$$\leq J C_1 e^{-C_2 n^2 \rho^{2J}},$$

which completes the proof. $\square$

## B   Addendum to experimental results

Figure 2: Tuning for $J$ in the simulated networks.

### B.1   Details of various graph statistics

The algebraic connectivity is the second smallest eigenvalue of the Laplacian. However, to make this metric free of the size of a graph, we use the second smallest eigenvalue of

the *normalized* Laplacian of the largest connected component of a graph. The need for using the largest connected component is that most real graphs without any preprocessing have isolated nodes, or small components. The global clustering coefficient measures the ratio of the number of triangles to the number of connected triplets. In contrast, the local clustering coefficient computes the average of the ratios of the number of triangles connected to a node and the number of tripes centered at that node. The distance distribution for $h$ hops essentially calculates the fraction of all pairs of nodes that are within shortest path or geodesic distance of $h$ hops. Essentially this metric calculates how far a pair of nodes are in a graph on average. The Pearson correlation coefficient of a graph measures the assortativity by computing the correlation coefficient between the degrees of the endpoints of the edges in the graph. Finally, the rich-club metric calculates the edge density of the subgraph induced by nodes with degree above a given threshold. For this metric we chose to use the 0.8-th quantile of the degree sequence of a graph.

## B.2    Tuning for $J$

In Figure 2, we plot the separation $(= (\lambda_\mathcal{K} - \lambda_{\mathcal{K}+1})/\lambda_{\mathcal{K}+1})$ found in the kernel matrix $\mathcal{K}$ against the value of $J$ used in NCLM in our simulation settings.