[Reviews · NeurIPS 2017]

Reviewer 1



The paper is interesting, and it has merit, but its limited level of novelty and achieved gain are hardly supporting acceptance. For what I checked, the proposed methods are correct, but the authors spend too much providing mathematical details rather than performing a more convincing comparison with other methods, which, at the moment, is showing only a fair improvement over similar algorithms.Furthermore, the paper is describing a clever combination of well known techniques, rather than introducing some novel ideas, and, overall, the role played by machine learning throughout the process is quite marginal. Finally, a few misprints occur in the main text, and the references are not homogeneously reported; in fact, some entries have all the authors with full name, others have initials, others have "et al." misplaced (see [3]) =============== The provided rebuttal addresses all the concern: overall rating modified accordingly.

Reviewer 2



Review Summary: This paper accomplishes a lot in just a few pages. This paper is generally well-written, but the first three sections in particular are very strong. The paper does a great job of balancing discussing previous work, offering theoretical results for their approach, and presenting experimental results on both generated and real data. The paper would benefit from just a touch of restructuring to add a conclusion and future work section as well as more details to some of their experimental methods and results. Summary of paper: This paper presents two versions of their framework for clustering networks: NCGE and NCLM. Each version tackles a slightly different variation of the problem of clustering networks. NCGE is applied when there is node correspondence between the networks being clustered, while NCLM is applied when there isn’t. It is impressive that this paper tackles not one but both of these situations, presenting strong experimental results and connecting their work to the theory of networks. This paper is superbly written in some sections, which makes the weaker sections almost jarring. For example, sections 2 and 3 do an excellent job of demonstrating how the current work builds on previous work. But then in sharp contrast lines 197-200 rely heavily the reader knowing the cited outside sources to grasp their point. Understandably (due to page lengths) one can not include detailed explanations for everything, but the authors may want to consider pruning in some places to allow themselves more consistency in their explanations. The last subsection (page 8, lines 299-315) is the most interesting and strongest part of that final section. It would be excellent if this part could be expanded and include more details about these real networks. Also the paper ends abruptly without a conclusion or discussion of potential future work. Perhaps with a bit of restructuring, both could be included. Section 3.2 was another strong moment of the paper, peppering both interesting results (lines 140-142) and forecasting future parts of the paper (lines 149-150). However, this section might be one to either condense or prune just a bit to gain back a few lines as to add details in other sections. A few formatting issues, questions, and/or recommendations are below: Page 1, lines 23,24,26, 35: Citations are out of order. Page 2, line 74: The presentation of the USVT acronym is inconsistent with the presentation of the NCGE and NCLM acronyms. Page 3, line 97: Citations are out of order. Page 4, line 128: Based on the earlier lines, it seems that g(A) = \hat{P} should be g(A_i) = \hat{P_i} Lines 159-160: In what sense does the parameter J count the effective number of eigenvalues? Page 5, lines 180-183: It appears that $dn$ is a number, but then in line 182, it seems that $dn()$ is a matrix function. Which is it? Lines 180-183: The use of three different D’s in this area make it a challenge to sort through what all the D’s mean. Perhaps consider using different letters here. Line 196: Citations are out of order. Page 6, line 247: How is clustering accuracy measured? Page 7, line 259: There appears to be an extra space (or two) before the citation [26] Lines 266-277: There appears to be a bit of inconsistency in the use of K. For example, in line 266, K is a similarity matrix and in line 268, K is the number of clusters. Lines 269-270, table 2, and line 284: What are GK3, GK4, and GK5? How are they different? Line 285: There is a space after eigenvectors before the comma.

Reviewer 3



Authors present a novel approach for clustering networks valued data with and without node correspondence. The approach is seemed to have the theoretical support for it’s consistency and tested through real and simulated data analyses. Overall, the paper is well written and I have some concerns on aspects outlined below. 1.Line 266: In learning t, how do you decide on the range to maximize the relative eigen-gap? I assume this is application specific? 2.Line 261: Thera are various networks based statistics and calculations also have various parameterizations. For example, how do you compute the clustering coefficient? You have a reference but there are various ways to define clustering coefficient using different measures and a brief discussion on this would enhance the paper. 3.Line 270: It’s not clear what is meant by “boosted it’s performance”. Do you mean computing time, or misclassification or convergence? 4.Line 280-281: This observation is counter intuitive relative the way you have defined similarity matrix and eigengap. It would be more interesting to see examine the behavior when the networks are heavily sparse. 5.There should be a section on discussion highlighting main results/contributions and also limitations of the approach. This is missing in the current version.